# Correlates of Loneliness and Social Isolation Among Korean Adults

**DOI:** 10.3390/healthcare13070806

**Published:** 2025-04-03

**Authors:** Inmyung Song

**Affiliations:** College of Nursing and Health, Kongju National University, Gongju 32588, Republic of Korea; inmyungs@kongju.ac.kr

**Keywords:** loneliness, social isolation, prevalence, correlate, Korea

## Abstract

**Background**: Loneliness is a public health concern. Despite the increasing attention paid to loneliness globally, knowledge regarding the condition in Korea is scarce. This study aims to examine the correlates of loneliness and social isolation among Korean adults. **Methods**: This cross-sectional study used a nationally representative sample of adults aged 30 years and older from the 2021 National Mental Health Survey of Korea (N = 4696). The survey used the six-item loneliness and social isolation scale (LSIS-6) to assess loneliness and social isolation between June and August, 2021. Two ordinary least squares regression models were conducted. Model 1 included socio-demographic variables as correlates of loneliness and social isolation. Model 2 added a range of mental health conditions, such as depressive disorder, anxiety disorder, alcohol-use disorder, nicotine-use disorders, and physical inactivity. Mental health was measured in relation to whether the participant had experienced each disorder in his/her lifetime. **Results**: A total of 34.4% of participants reported that they felt lonely at least occasionally. After all adjustments, the experience of mental health conditions was associated with an increase in the LSIS-6 score (B = 2.32 for depressive disorder, B = 0.59 for anxiety disorder, B = 0.36 for both alcohol-use disorder and nicotine-use disorder; *p* < 0.01). In addition, greater loneliness and social isolation were associated with male gender, older age, a lower educational level, non-married status, a lower household income, having a smaller number of children, having a greater number of chronic conditions, and taking less frequent walks per week. **Conclusions**: In conclusion, loneliness is prevalent among Korean adults. Loneliness and social isolation correlated significantly with socio-demographic characteristics and the experience of mental health conditions.

## 1. Introduction

Longitudinal studies have shown that loneliness and social isolation are associated with poor health outcomes later on [1], a diagnosis of chronic illness [2], and increased risk of all-cause mortality [3,4]. The range of outcomes was so wide that they included not only negative physical health outcomes but also mental health outcomes, cognitive function, quality of life [5,6], and greater healthcare utilization [7]. More than anything, loneliness is prevalent. A meta-analysis revealed that the prevalence of loneliness ranged from 2.7% to 9.6% for middle-aged adults and from 5.2% to 21.3% for older adults in European countries prior to the coronavirus disease 2019 (COVID-19) pandemic [8]. Against this backdrop, loneliness has been recognized as a growing public health problem and this was critically assessed even before the pandemic [9]. While information is scare in Korea, it was shown that the prevalence of social isolation and loneliness was 17.8% and 4.1%, respectively, based on a sample of Koreans from three cities in 2019 [10].

Then, the COVID-19 pandemic hit the globe, which led to the imposition of social-distancing measures and lockdowns. Consequently, the problem of loneliness appears to have grown larger across the world [11,12]. The prevalence of loneliness among adults was 27.0% in the U.K. [13] and 34.7% in Canada [12]. In Korea, 20.2% of adults reported feeling lonely at least one day a week in 2021 [14]. Loneliness is so prevalent in modern society that one could call it an “epidemic” [12,15]. Addressing the growing concern regarding this epidemic would call for the identification of its risk factors and correlates.

So far, loneliness and social isolation have been associated with socio-demographic factors, such as age [16,17], female gender [18], a lower educational level [18], non-married status [14,19], living alone [18,19], unemployment [20], a poor income [18], and having a smaller number of children [21]. Another important category of risk factors for loneliness was health status [22]. In particular, the number of chronic illnesses predicted loneliness among older adults in the United States [19]. In addition to physical health status, poor mental health also predicted loneliness [16,18]. For example, research suggests that COVID-19-related anxiety was associated with increased loneliness [23]. There is also evidence that loneliness during the pandemic correlates with a depressive mood and anxiety [24,25]. Depression and anxiety are closely related to each other; however, loneliness is recognized as a distinct construct that is distant from the two mental disorders [26]. Therefore, it is worthy of examining the association between loneliness and mental conditions.

Among other conditions linked to loneliness were substance use problems, such as alcohol dependency [27]. However, there exists contradictory evidence that loneliness is positively associated with alcohol consumption [28] but negatively associated with the frequency of drinking [29]. Similarly, the association between smoking and loneliness has been studied but the findings are inconclusive [30]. A recent study based on older adults in the U.K. suggests that smoking is associated with higher levels of social disengagement and loneliness [31]. On the other hand, healthy behavior, such as conducting moderate-to-high physical activity, was shown to be a protective factor for loneliness and social isolation among older adults in the U.S. [32]. Nevertheless, physical inactivity did not predict loneliness among older Canadians during the pandemic [33]. Taken together, the association between loneliness and physical activity was also inconclusive [34].

While studies on the prevalence of loneliness and its risk factors are abundant worldwide, understanding of these issues in the Korean population remains limited. Investigating this specific population could be meaningful, as the measures taken during the pandemic, people’s responses, and the activities that provide mental satisfaction differ between Korea and other cultures [35,36]. Korea has one of the highest suicide rates globally, with social isolation cited as a contributing factor [37]. Furthermore, the population ranks among the least happy in the world [38]. Under the circumstances, examining the psychological well-being of the population—particularly in in regard to loneliness and social isolation—is crucial. However, existing studies in Korea are either based on a sample from a few select cities or rely on a single-item measure of loneliness [10,14,25].

Loneliness is defined as the unpleasant feeling that individuals experience when they are not satisfied with the quality and quantity of social relations [39]. In that sense, loneliness is a subjective assessment, whereas social isolation is largely concerned with the objective condition of social disconnectedness [40]. While these concepts are distinct and only weakly correlated [41], they are often measured together using comprehensive instruments that require numerous question items, which can limit their practicality [42]. Consequently, some researchers have highlighted the need for a concise, practical scale to assess and monitor loneliness and social isolation in large-scale surveys [43]. Furthermore, early loneliness scales were primarily developed for Western populations, but in recent years, more tools have been designed for diverse populations [44,45]. In response to this need, a simplified scale was developed and implemented in a nationwide survey in Korea to screen for loneliness and social isolation in community settings [46,47].

Within this context, this study aims to examine the conditions of loneliness and social isolation and their relationship with socio-demographic characteristics and health status using a nationally representative sample of adults in Korea during the COVID-19 pandemic. In addition, this study will utilize a more comprehensive measure of loneliness and social isolation, specifically developed for Korean adults [46]. Furthermore, this study will re-examine the correlates that have produced mixed findings in previous studies, such as disorders associated with alcohol use [28,29], smoking [30,31], and physical inactivity [32,33]. Understanding the correlates of loneliness and social isolation will inform the development of preventive interventions.

## 2. Materials and Methods

### 2.1. Data

This study used data from the National Mental Health Survey of Korea (NMHSK) in 2021. The National Mental Health Center of Korea in collaboration with the Ministry of Health and Welfare has conducted the survey every five years since 2001 in an effort to inform policies to improve mental health care [47]. The latest wave of the NMHSK was conducted between June and August, 2021, while social-distancing measures were still in place during the COVID-19 pandemic [48]. To improve the representativeness of the sample, the NMHSK used stratifications based on bigger (city vs. province) and smaller (dong vs. eup) administrative districts. A sample of 5511 participants aged 18–79 years were selected and interviewed for the 2021 survey. Trained interviewers conducted one-on-one interviews using a tablet PC that was installed with a survey program. Survey participants also self-administered some questionnaire items, such as those regarding loneliness, social isolation, and health behaviors. This current study analyzed only the adults aged 30 years and older (n = 4696) and excluded people in their 20s who are likely to be still single, in education, and unemployed. The reason for their exclusion is that young adults in their 20s differ significantly from older age groups in terms of key characteristics such as marital status, employment, and family responsibilities. Moreover, they are often still cared for by their parents and do not typically deal with the work and family issues that older age groups face. These differences are substantial enough to warrant a closer examination in a separate study.

### 2.2. Loneliness and Social Isolation Measure

A number of measures of loneliness have been developed but no universal measure exists [49]. Hwang et al. (2021) developed the six-item loneliness and social isolation scale (LSIS-6) with a goal of creating a convenient screening tool to be used for Koreans in the community setting [46]. The newly developed LSIS-6 was used in the 2021 NMHSK [47]. As a comprehensive multi-dimensional measure, the LSIS-6 has a three-factor structure comprising social support, social networks, and loneliness. The LSIS-6 is composed of the following four statements and two questions describing the status of loneliness and social isolation in the past month: (1) I feel lonely; (2) I feel isolated; (3) I can comfortably rely on friends and family; (4) There are people who can help me with everyday matters; (5) With how many people are you close enough to meet in person at least once a month or contact at least once a week (including family members, relatives, and friends)?; and (6) On average, how many minutes per day do you spend speaking with friends and family (through phone call, texting, Kakaotalk, and other messenger apps)?

Agreement with each statement was rated on a 4-point Likert scale; strongly disagree = 3 points, somewhat disagree = 2 points, somewhat agree = 1 point, and strongly agree = 0 points. Responses to the question on the number of people that the respondents contact were measured on a scale of 0–3: 0 persons = 3 points, 1 to 2 persons = 2 points, 3 to 6 persons = 1 point, and 7 or more persons = 0 points. Similarly, responses regarding the minutes spent with friends and family were measured as follows: none = 3 points, 15 min or less = 2 points, 16 to 59 min = 1 point, and 1 h or more = 0 points. Responses to the six items were summed to create a summary score ranging from 0 to 18. Internal consistency for all items of the LSIS-6 was good (Cronbach’s α = 0.774) and the LSIS-6 was significantly correlated with other well-validated measures of social isolation and loneliness, such as the UCLA Loneliness Scale 8 and Lubben Social Network Scale 6 [46].

In addition to the LSIS-6, a single-item measure of loneliness was also used in the 2021 NMHSK. The question asked how often the participant feels lonely. To that question, the participant was to select rarely, occasionally, or frequently.

### 2.3. Socio-Demographic Factors

This study examined socio-demographic characteristics as potential correlates of loneliness. The characteristics comprised sex, age, marital status, educational level, living arrangement, number of children, employment status, and household income level. Age was grouped into 30–39, 40–49, 50–59, 60–69, and ≥70 years. Educational level was categorized into ≤elementary school, middle school, high school, and ≥college. Marital status was categorized into married, widowed, separated/divorced, and single. Living arrangement was dichotomized into living alone and living with others. The number of children was categorized as 0, 1, 2, and ≥3. Employment status was categorized as employed and not employed. Household income level was categorized as <median income and ≥median income.

### 2.4. Health Status and Health Behavior

This study used two dimensions of health status, namely physical and mental. Physical health status was operationalized as the number of chronic conditions, as it was in the literature [33]. The number of chronic conditions was obtained by summing the number of self-reported chronic conditions collected in the NMHSK (hypertension, dyslipidemia, stroke, myocardial infarction, angina, diabetes mellitus, and cancer).

Mental health status was assessed in terms of depressive disorder and anxiety disorder. In the NMHSK, mental health was assessed with an adaptation of the Composite International Diagnostic Interview (CIDI). The CIDI was initially developed, at the request of the World Health Organization, as an instrument to assess psychiatric disorders administrable by lay interviewers [50]. The CIDI was modified to better suit the sociocultural context of Korea [51]. The Korean version was used in the NMHSK to diagnose a range of mental health disorders in the community setting [47]. The NMHSK also assessed alcohol-use disorder and nicotine-use disorder. The former was defined as a disorder associated with alcohol use including alcohol abuse and dependency. The latter was defined as a disorder associated with nicotine use including nicotine withdrawal and nicotine dependency. The NMHSK provides data on the experience of each disorder based on a set of questions, indicating whether or not the participant has experienced each disorder in his/her lifetime. Health behavior was defined as the frequency of 10 min walks taken per week and categorized as 0–3, 4–6, and 7 times.

### 2.5. Statistical Analysis

Descriptive statistics were obtained as the frequency of respondents in the sample and the percentage in the weighted sample, according to socio-demographic characteristics, health status, and behavior. The prevalence of loneliness was calculated as the proportion of the participants who answered occasionally or frequently to the single-item measure of loneliness. Sampling weights were applied in all analyses to take account of the complex survey design of the NMHSK. The mean LSIS-6 scores were calculated for each category of socio-demographic characteristics. The *t*-test and analysis of variance (ANOVA) were used to test if there was a difference in the mean LSIS-6 scores between and among categories. The Tukey–Kramer method was used for post hoc pairwise multiple comparisons.

Ordinary least squares (OLS) regression was used to determine the correlates of loneliness and social isolation. The dependent variable was the LSIS-6 score, which was used as a continuous variable for two reasons. First, so far, there is no validated cut-off point for the LSIS-6 scores to dichotomize loneliness and social isolation. Second, other loneliness measures, most notably the three-item UCLA Loneliness Scale, have also been treated as a continuous variable [17] and OLS models were applied to analyze the data [11,31]. This current study implemented two OLS models in steps with covariates entered sequentially. Model 1 included socio-demographic characteristics (sex, age, educational level, marital status, living arrangement, number of children, employment status, and household income). Model 2 added all health-related variables (number of chronic conditions, experience of depressive disorder, anxiety disorder, alcohol-use disorder, nicotine-use disorder, and frequency of 10 min walks per week). Collinearity was checked and turned out not to be an issue based on the examination of tolerance values, as all variables had values greater than 0.1.

All statistical analyses were performed by using SAS version 9.4 (Cary, NC, USA). The Institutional Review Board of Kongju National University approved the study protocol and waived the requirement for informed consent (reference No. KNU_IRB_2023-018).

## 3. Results

In the weighted sample, 50.1% were female and 25.5% were in their 50s (Table 1). A total of 41.0% respondents attained college and higher education; 76.4% were married and 42.8% lived alone; 49.9% had two children and 14.2% had three or more children; 72.7% respondents were employed; 10.8% had two or more chronic conditions; 8.1% and 9.4% reported experiencing depressive disorder and anxiety disorder, respectively; 12.0% and 10.2% experienced alcohol-use disorder and nicotine-use disorder, respectively; and 31.2% felt lonely occasionally and 3.2% felt lonely frequently.

The mean LSIS-6 score was slightly higher for men than for women (*p* < 0.05) and increased with advanced age (*p* < 0.001) (Table 2). College-educated and married people were less lonely and socially isolated than people with lower educational attainment and other marital statuses, respectively (*p* < 0.001). People who lived alone and the unemployed felt lonelier and more socially isolated than those who lived together and the employed, respectively (*p* < 0.001). People with a lower income were lonelier and more socially isolated (*p* < 0.001). The mean LSIS-6 score increased with the number of chronic conditions as well as with the experience of depressive, anxiety, alcohol-use, and nicotine-use disorders (*p* < 0.001). The mean LSIS-6 score decreased with the frequency of 10 min walks per week (*p* < 0.001).

In regression models, men were lonelier and more socially isolated than women (*p* < 0.001) and loneliness and social isolation were associated with advanced age (*p* < 0.01 for 40–49 years; *p* < 0.001 for 50 years and older) (Table 3). People who did not obtain a college education were lonelier and more socially isolated than those who were college educated (*p* < 0.001). People who were widowed, divorced/separated, or single were lonelier and more socially isolated than those who were married (*p* < 0.001). People who had no children or only one child were lonelier and more socially isolated than those who had three or more children (*p* < 0.001). The unemployed were lonelier and more socially isolated than the employed (*p* < 0.01). Low household income was a significant predictor of loneliness and social isolation (*p* < 0.001). After controlling for socio-demographic variables, health status and behavior were significant predictors of loneliness and social isolation in Model 2. People with two or more chronic conditions were lonelier and more socially isolated than people with none (B = 0.73, *p* < 0.001). Depressive disorder and anxiety disorder were significantly associated with loneliness and social isolation (B = 2.32 and B = 0.59, respectively, *p* < 0.001). Loneliness and social isolation were predicted by the experience of either alcohol- or nicotine-use disorder (B = 0.36, *p* < 0.01) and having less frequent walks per week (B = 0.67 for 0–3 times; B = 0.53 for 4–6 times, *p* < 0.001). Model 1 explained 14% of the variance in loneliness and social isolation, and adding health variables in Model 2 explained additional 8%.

## 4. Discussion

This population-based study estimates that one in three adults in Korea felt lonely at least occasionally during the COVID-19 pandemic. The prevalence of loneliness obtained using a single-item measure of loneliness in this present study is higher than previous estimates (17.8–20.2%) [10,14]. However, the prevalence (34.4%) in Korea is comparable to that in Canada (34.7%) and the U.K. (27.0%) [12,13].

Among the socio-demographic characteristics examined in this study, gender was a significant predictor of loneliness and social isolation measured in the LSIS-6 score. In particular, men were lonelier and more socially isolated than women in Korea. This finding is consistent with that of a meta-analysis [52], but is contrary to the recent findings in the U.K. [53], Germany [54], and Canada [12]. The difference might be due, in part, to a potential influence of the measure used. The LSIS-6 used in this study is designed to capture the social isolation dimension in addition to loneliness, which might have contributed to the difference. It is likely that men are more socially isolated but more hesitant to report loneliness than women [10].

Age was another demographic variable that had a less than straightforward relationship with loneliness and social isolation. In this present study, older adults were at a greater risk of loneliness and social isolation. The literature also shows that loneliness increased with advanced age across nations [55]. However, there is counterevidence according to studies in the U.K. [11,13], Canada [12], and the U.S. [56]. A study even demonstrated a U-shaped relationship between age and loneliness [16]. Regardless of the direction of the relationship, age appears to be an important predictor of loneliness and social isolation across countries.

This present study showed that non-married individuals were lonelier and more socially isolated than married individuals and that a lower number of children was negatively associated with loneliness and social isolation among adults. These findings are consistent with those in the U.S. and Germany [54,56]. Certain demographic subgroups, such as women and older adults, appear to have benefited from having children in terms of lower social loneliness [57]. In addition to marital status and having children, another living status variable, namely living with others, was shown to have an influence on loneliness [12,54]. In particular, living with a greater number of adults was a protective factor for loneliness in the U.K. during the COVID-19 pandemic [13,53]. However, despite the ample evidence in other countries, living alone was not a significant predictor of loneliness and social isolation in the present study. One possible explanation for this difference is that, unlike in some other high-income countries, most people in Korea live in high-rise apartments in densely populated urban areas. Consequently, living alone does not necessarily equate to physical or social isolation. It is likely that Koreans living alone may not necessarily feel lonely and socially isolated and that they can meet or contact with others on a regular basis. Whether there is really a difference in the role of living alone in the subjective assessment of loneliness and social isolation across countries should be examined with a uniform measure of loneliness and social isolation.

In this study, adults who received less than a college education were lonelier and more socially isolated than those with a college education. Research in other countries has also showed that attaining a higher level of education was independently protective of loneliness among older adults [58,59]. This relationship may have emerged, in part, because education may help to build social skills and other abilities to respond to loneliness [60]. Another possible mechanism is that educational attainment has an indirect influence on loneliness mediated through less neuroticism and stress [59]. Besides educational level, two economic indicators were significant correlates of loneliness and social isolation in this present study. Unemployed adults and adults with a lower family income were lonelier and more socially isolated than their counterparts. Likewise, a low income and economic inactivity were associated with an increased risk of loneliness during the COVID-19 pandemic in the U.K. [53]. However, it should be recognized that there is the potential for a bi-directional relationship between loneliness and unemployment or income level [61].

In this current study, the number of chronic conditions was a significant predictor of loneliness and social isolation, which was consistent with a previous finding [19]. Research indicates that disease burden from chronic illnesses increases psychological distress and the risk of social isolation and loneliness [62]. Moreover, experience of depressive disorder was the strongest predictor of loneliness and social isolation in this current study. Consistent with this finding, scores meeting the clinical criteria for depression and having more depressive symptoms were associated with an increased risk of loneliness in the U.K. [13,63]. Although not as strong a predictor as depressive disorder, the experience of anxiety disorder was a significant predictor of loneliness and social isolation in this present study, consistent with the finding in the U.S. [56]. While mental conditions such as depression and anxiety were shown to predict loneliness, research supports that the relationship can be bi-directional [64].

In this study, the experience of either alcohol-use disorder or nicotine-use disorder were significant predictors of loneliness and social isolation. Similarly, earlier studies suggest that problem drinking and the initiation of smoking were associated with a greater risk of loneliness during the COVID-19 pandemic [12,33,65]. Like with the cases of depression and anxiety, smoking and alcohol-use disorder also appear to have a bi-directional relationship with loneliness [31,65]. Despite the lack of firm evidence in other countries [33,34], taking regular walks per week was associated with a decrease in loneliness and social isolation in this present study. However, one should recognize the bi-directionality of the negative association between physical activity and loneliness [66].

Using a nationally representative sample of Korean adults, this study provides an update on the prevalence of loneliness among Korean adults, which is shown to be higher than previously estimated. Moreover, this study examined the correlates of loneliness and social isolation. For this analysis, data collected on a multi-item LSIS was utilized, which allowed for capturing the varied dimensions of loneliness and social isolation. Despite these strengths, this study has the following limitations: First, this study used a cross-sectional design and, therefore, causal inferences may not be drawn from the findings, nor did the current findings support the bi-directionality of the relationships that have been suggested in previous studies [31,65]. Second, this study relied upon self-reporting, which may be susceptible to recall bias. Third, while the LSIS-6 was used to conveniently monitor loneliness and social isolation in a nationwide survey, combining the two distinct concepts in a six-item measure may be too simplistic to capture the varied dimensions of the complex conditions.

Notwithstanding these limitations, the findings of this study will expand the knowledge base of loneliness and social isolation in the Korean population and aid in the development of interventions aiming to reduce the growing public health problem. First, this study showed that loneliness is a highly prevalent condition in Korea, emphasizing that efforts should be directed towards addressing the issue from the public health standpoint. Second, this study identified highly vulnerable segments of the population, such as men, older age groups, and people with few children. To maximize the effect, the policy interventions to reduce loneliness and social isolation should be targeted at these groups, particularly those with multiple chronic conditions and/or health-behavior issues related to nicotine and alcohol use. Third, the findings highlighted the correlates of loneliness and social isolation, such as some mental health conditions and physical inactivity. Care providers should recognize these correlates as potential areas of clinical and behavioral intervention at the practice level or in community settings.

## 5. Conclusions

Based on a nationally representative sample, this study showed that loneliness is prevalent among Korean adults aged 30 years and older. Loneliness and social isolation were associated with male gender, older age, a lower educational level, non-married status, a lower household income, having a smaller number of children, having a greater number of chronic conditions, experience of mental health conditions, and taking less frequent walks per week. In particular, depressive disorder was the strongest predictor of loneliness and social isolation. Identifying the risk factors for loneliness and social isolation can point to potential targets for public health intervention.

## Figures and Tables

**Table 1 healthcare-13-00806-t001:** Characteristics of study subjects (n = 4696).

Variable	Category	No. of Respondents	Weighted %
Sex	Male	2336	49.9
	Female	2360	50.1
Age group, years	30–39	838	18.4
	40–49	1111	24.3
	50–59	1231	25.5
	60–69	1053	20.9
	≥70	463	11.0
Mean age (±SD)	52.7 (±12.4) years		
Educational level	≤Elementary school	356	7.8
	Middle school	478	10.3
	High school	1921	40.1
	≥College	1894	41.9
Marital status	Married	3549	76.4
	Widowed	308	6.8
	Separated/divorced	336	5.6
	Single	503	11.3
Living arrangement	Living alone	2300	42.8
	Living together	2396	57.2
No. of children	0	762	16.2
	1	926	19.6
	2	2322	49.9
	≥3	686	14.2
Employment status	Employed	3456	72.7
	Unemployed	1240	27.3
Household income level	<Median income	2279	47.1
	≥Median income	2396	52.9
No. of chronic conditions	0	3320	71.0
	1	847	18.2
	≥2	529	10.8
Depressive disorder	Yes	358	8.1
	No	4338	91.9
Anxiety disorder	Yes	441	9.4
	No	4255	90.6
Alcohol-use disorder	Yes	563	12.0
	No	4133	88.0
Nicotine-use disorder	Yes	458	10.2
	No	4238	89.8
Frequency of 10 min walks per week	0–3 times	1550	32.0
	4–6 times	1604	35.1
	7 times	1542	32.8
Loneliness	Rarely	3076	65.6
	Occasionally	1474	31.2
	Frequently	146	3.2
Mean LSIS-6 score (±SD)	6.0 (±2.8)		

% (weighted) is the estimated percentage in the population. Abbreviation: LSIS-6, the 6-item loneliness and social isolation scale.

**Table 2 healthcare-13-00806-t002:** Mean loneliness and social isolation scale-6 (LSIS-6) score by characteristics (n = 4696).

Variable	Category	Mean	SE	*p*-Value	95% Confidence Interval *
Sex	Male	6.17	0.08	0.015	
	Female	5.97	0.08		
Age group, years	30–39	5.43	0.11	<0.001	
	40–49	5.58	0.11		−0.13, 0.44
	50–59	6.00	0.10		0.31, 0.84 ***
	60–69	6.67	0.11		0.96, 1.53 ***
	≥70	7.20	0.15		1.40, 2.14 ***
Educational level	≤Elementary school	7.30	0.18	<0.001	1.51, 2.30 ***
	Middle school	7.31	0.16		1.58, 2.25 ***
	High school	6.21	0.09		0.61, 1.01 ***
	≥College	5.40	0.08		
Marital status	Married	5.68	0.07	<0.001	
	Widowed	7.75	0.19		1.70, 2.45 ***
	Separated/divorced	8.22	0.22		2.10, 2.98 ***
	Single	6.64	0.16		0.65, 1.27 ***
Living arrangement	Living alone	6.66	0.09	<0.001	
	Living together	5.62	0.08		
No. of children	0	6.50	0.13	<0.001	−0.19, 0.51
	1	5.87	0.12		−0.80, −0.14 **
	2	5.92	0.08		−0.69, −0.14 **
	≥3	6.34	0.14		
Employment status	Employed	5.87	0.07	<0.001	
	Unemployed	6.59	0.12		
Household income level	<Median income	6.46	0.09	<0.001	
	≥Median income	5.72	0.08		
No. of chronic conditions	0	5.73	0.07	<0.001	
	1	6.50	0.12		0.54, 1.00 ***
	≥2	7.52	0.17		1.45, 2.13 ***
Depressive disorder	Yes	8.73	0.24	<0.001	
	No	5.83	0.07		
Anxiety disorder	Yes	7.26	0.19	<0.001	
	No	5.95	0.07		
Alcohol-use disorder	Yes	6.89	0.18	<0.001	
	No	5.96	0.07		
Nicotine-use disorder	Yes	6.75	0.19	<0.001	
	No	5.99	0.07		
Frequency of 10 min walks per week	0–3 times	6.32	0.12	<0.001	0.29, 0.87 ***
	4–6 times	6.14	0.10		0.15, 0.66 **
	7 times	5.74	0.10		

* The Tukey–Kramer method was used for pairwise post hoc comparisons and to obtain 95% confidence intervals for the mean difference. *** *p* < 0.001, ** *p* < 0.01. The *p*-values were calculated using ANOVA or *t*-tests.

**Table 3 healthcare-13-00806-t003:** Results of multiple regression analyses on the 6-item loneliness and social isolation scale score.

		Model 1			Model 2		
Variable (Reference)	Category	B	SE	*p*-Value	B	SE	*p*-Value
Sex (female)	Male	0.41	0.09	<0.001	0.35	0.09	<0.001
Age group (30–39), years	40–49	0.42	0.13	0.001	0.40	0.12	0.001
	50–59	0.91	0.15	<0.001	0.75	0.14	<0.001
	60–69	1.09	0.17	<0.001	0.82	0.17	<0.001
	≥70	1.20	0.22	<0.001	0.97	0.22	<0.001
Educational level (≥college)	≤Elementary school	0.91	0.21	<0.001	0.92	0.20	<0.001
	Middle school	1.17	0.17	<0.001	1.19	0.16	<0.001
	High school	0.55	0.10	<0.001	0.53	0.09	<0.001
Marital status (married)	Widowed	1.38	0.18	<0.001	1.07	0.17	<0.001
	Separated/divorced	2.14	0.18	<0.001	1.71	0.17	<0.001
	Single	0.93	0.21	<0.001	0.88	0.20	<0.001
Living arrangement (living together)	Living alone	−0.15	0.10	0.149	−0.06	0.10	0.515
No. of children (≥3)	0	1.24	0.23	<0.001	1.11	0.22	<0.001
	1	0.65	0.15	<0.001	0.60	0.14	<0.001
	2	0.24	0.13	0.053	0.22	0.12	0.072
Employment status (employed)	Unemployed	0.48	0.10	<0.001	0.32	0.10	0.001
Household income level (≥median income)	<Median income	0.40	0.09	<0.001	0.34	0.08	<0.001
No. of chronic conditions (0)	1				0.23	0.11	0.032
	≥2				0.73	0.14	<0.001
Depressive disorder (no)	Yes				2.32	0.14	<0.001
Anxiety disorder (no)	Yes				0.59	0.13	<0.001
Alcohol-use disorder (no)	Yes				0.36	0.12	0.003
Nicotine-use disorder (no)	Yes				0.36	0.13	0.006
Frequency of 10 min walks per week	0–3 times				0.67	0.09	<0.001
	4–6 times				0.53	0.09	<0.001
Constant		3.75			3.18		
No. of observations		4630			4630		
R-square		0.14			0.22		

## Data Availability

The data used for this study are owned by a third party. A request for the data used for this study can be made via https://mhs.ncmh.go.kr/eng/ (accessed on 3 May 2023).

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
