# Peer review of "Correlates of Loneliness and Social Isolation Among Korean Adults"

_healthcare, 2025, doi:10.3390/healthcare13070806_

Round 1
Reviewer 1 Report
Comments and Suggestions for Authors
Review:
I appreciate the opportunity review the manuscript “Correlates of loneliness and social isolation among Korean adults.” Overall, I find the findings to be robust and understandable. However, there are several major issues that limit the impact of the findings. To summarize, I recognize that there is relatively little in sociodemographic correlates of loneliness in Korea specifically; however, the findings seem to replicate what is already established across numerous national contexts. I believe that the manuscript would benefit from either defining more clearly what is unique about the Korean subsample and/or expanding the methods to provide more unique insights into the mechanisms for why such correlates are observed.
The introduction section is a bit disjointed and needs a substantial revision to more clearly identify the motivation and foundation of this study. For example, what scholarly gap is this study contributing to? I was unsure if the problem is that loneliness is a particularly unique problem in Korea or whether the issue was more of a methodological issue in measurements of loneliness. This needs to be more clearly specified.
Generally in the introduction section, I caution the authors from using causal language. For example, the literature more establishes a positive correlation between smoking and social disengagement but not exactly that it “increases” disengagement and loneliness. It is equally plausible that those who are isolated and disengaged smoke rather than the other way around.
While there are numerous studies on isolation and loneliness among Korean older adults, the introduction did not sufficiently establish how the existing measures of loneliness are not adequate. I do not necessarily disagree with the author(s) on this, but more needs to be stated regarding the validity issue of loneliness measures.
Methods:
I question the logic of including such a large range of ages (from 30 to 79) in this data. While I agree that people in their 20s are dissimilar to older adult profiles, I would also argue that people in their 30s are equally dissimilar. Generally, people in their 30s are not far removed from university educations, have young children, and are more likely to be dealing with work-family balance issues that many older adults do not experience. I believe that this sample either needs to be justified further or keep the sample closer to retirement age (i.e. 55+).
The authors should clarify item 6 (minutes per day speaking with friends/family). Is that on a Likert scale measure rating agreement? If so, then what were the respondents agreeing to?
Please include the summary statistics of key measures (i.e. means, standard deviations of loneliness and social isolation measure).
I disagree with the entanglement of loneliness and social isolation as the same measure. For example, it is plausible for a person to be lonely but not socially isolated and vice versa. For example, it does not seem appropriate to include item 4, 5, and 6 along with the first 3 items, which more rate the person’s attitudes. If you measure items 1, 2, and 3 as loneliness and 4, 5 and 6 as isolation, would you get similar results? I believe the paper would be strengthened by separating the two concepts and testing for similarities/differences.
Minor issues:
References should be in order
On line 54, the sentence “Research suggests that COVID-19 anxiety had a negative in fluence in loneliness” is unclear as it reads suggesting that that anxiety reduced loneliness.
Author Response
Comments 1: I appreciate the opportunity review the manuscript “Correlates of loneliness and social isolation among Korean adults.” Overall, I find the findings to be robust and understandable. However, there are several major issues that limit the impact of the findings. To summarize, I recognize that there is relatively little in sociodemographic correlates of loneliness in Korea specifically; however, the findings seem to replicate what is already established across numerous national contexts. I believe that the manuscript would benefit from either defining more clearly what is unique about the Korean subsample and/or expanding the methods to provide more unique insights into the mechanisms for why such correlates are observed.
Response 1: I appreciate the thoughtful feedback and have done my best to address each comment below. However, if any of my responses are unsatisfactory, please let me know.
Comments 2: The introduction section is a bit disjointed and needs a substantial revision to more clearly identify the motivation and foundation of this study. For example, what scholarly gap is this study contributing to? I was unsure if the problem is that loneliness is a particularly unique problem in Korea or whether the issue was more of a methodological issue in measurements of loneliness. This needs to be more clearly specified.
Response 2: I understand that loneliness is not unique to Korea, but the country faces a unique challenge such as world’s highest suicide rate and extremely low happiness rate, points that were emphasized in the revised paper. Nonetheless, the population had not been assessed for loneliness and social isolation using a comprehensive measure in a nationwide survey until the survey used in this study was conducted. In that sense, both factors play a role. I strengthened the manuscript by highlighting key points in certain parts of the introduction. Additionally, I removed redundancies and extensively revised the paper to improve its overall flow. (the last three paragraphs of the introduction section).
Comments 3: Generally in the introduction section, I caution the authors from using causal language. For example, the literature more establishes a positive correlation between smoking and social disengagement but not exactly that it “increases” disengagement and loneliness. It is equally plausible that those who are isolated and disengaged smoke rather than the other way around.
Response 3: I revised the introduction and other sections so as not to suggest causality, as in “ … smoking is associated with higher levels of social disengagement and loneliness (lines 67-69) and “Research suggests that COVID-19 anxiety was associated with increased loneliness. (lines 56-57).”
Comments 4: While there are numerous studies on isolation and loneliness among Korean older adults, the introduction did not sufficiently establish how the existing measures of loneliness are not adequate. I do not necessarily disagree with the author(s) on this, but more needs to be stated regarding the validity issue of loneliness measures.
Response 4: I added the following to address some issues of loneliness measures: While these concepts are distinct and only weakly correlated, they are often measured together using comprehensive instruments that require numerous question items, which can reduce their practicality…… Furthermore, early loneliness scales were primarily developed for Western populations, but in recent years, more tools have been designed for diverse populations. (lines 86-92)
Methods:
Comments 5: I question the logic of including such a large range of ages (from 30 to 79) in this data. While I agree that people in their 20s are dissimilar to older adult profiles, I would also argue that people in their 30s are equally dissimilar. Generally, people in their 30s are not far removed from university educations, have young children, and are more likely to be dealing with work-family balance issues that many older adults do not experience. I believe that this sample either needs to be justified further or keep the sample closer to retirement age (i.e. 55+).
Response 5: I agree that each age group has its own issues and concerns. However, limiting the analysis to a small segment of the population will not provide a comprehensive view of the extent of loneliness and social isolation across the entire population. Perhaps a future study will explore the unique challenges faced by different age groups. One of the goals of this study is to examine the prevalence of loneliness and social isolation in a broader context. Therefore, rather than focusing on one specific age group, I included all age groups. The reason for excluding people in their 20s is that, much like adolescents in Korea, they are often still under the care of their parents and typically do not establish their own households. I justified the sample by including the following; The reason for their exclusion is that young adults in their 20s differ significantly from older age groups in terms of key characteristics such as marital status, employment, and family responsibilities. Moreover, they are often still cared for by their parents and do not typically deal with the work and family issues that older age groups face. These differences are substantial enough to warrant a closer examination in a separate study. (in section 2.1)
Comments 6: The authors should clarify item 6 (minutes per day speaking with friends/family). Is that on a Likert scale measure rating agreement? If so, then what were the respondents agreeing to?
Response 6: Thank you for pointing out the omission. I added how the question and another were scored in the revised manuscript. (lines 141-145)
Comments 7: Please include the summary statistics of key measures (i.e. means, standard deviations of loneliness and social isolation measure).
Response 7: I included the means and standard deviations of age and the LSIS-6 in Table 1.
Comments 8: I disagree with the entanglement of loneliness and social isolation as the same measure. For example, it is plausible for a person to be lonely but not socially isolated and vice versa. For example, it does not seem appropriate to include item 4, 5, and 6 along with the first 3 items, which more rate the person’s attitudes. If you measure items 1, 2, and 3 as loneliness and 4, 5 and 6 as isolation, would you get similar results? I believe the paper would be strengthened by separating the two concepts and testing for similarities/differences.
Response 8: I sincerely appreciate your feedback. While I acknowledge that loneliness and social isolation can be examined as distinct concepts, the rationale for utilizing the LSIS-6 in the 2021 National Mental Health Survey of Korea is to employ a comprehensive screening instrument for the population. Separating the tool into two distinct concepts at my discretion is possible; however, it may be challenging to justify, given that the LSIS-6 is designed to encompass three dimensions: social support, social networks, and loneliness. Additionally, the tool has demonstrated strong internal consistency. To examine loneliness and social isolation separately, it may be more appropriate to utilize distinct instruments tailored to each concept.
Minor issues:
Comments 9: References should be in order.
Response 9: I checked that references are in order in the revised paper.
Comments 10: On line 54, the sentence “Research suggests that COVID-19 anxiety had a negative in fluence in loneliness” is unclear as it reads suggesting that that anxiety reduced loneliness.
Response 10: I revised the sentence to ensure it does not suggest causality; Research suggests that COVID-19 anxiety was associated with increased loneliness. (lines 56-57)
Reviewer 2 Report
Comments and Suggestions for Authors
Thank you for inviting me to review this paper. This cross-sectional study examines the factors associated with loneliness and social isolation among Korean adults. Given the growing social concerns on loneliness and social isolation both in Korea and globally, this study is timely and valuable. This paper is very well-written and based on a sound methodology. Particularly, the use of a nationally representative sample and a validated multi-item measurement (LSIS-6) enhances the generalizability of the findings. I appreciate the author' efforts and would like to offer a few minor suggestions for improvement.
Methods
The Method section provides a clear and reasonable explanation of the sampling procedure, measurement, and statistical analysis.
- Section 2.1: It would be helpful to briefly describe the COVID-19 situation in Korea between June and August 2021, along with the relevant social distancing policies, to provide context.
- Lines 190–191: How was collinearity checked? If the variance inflation factor (VIF) was used, please specify its cutoff value for ruling out multicollinearity.
Results
- Table 2: Please include a table footnote indicating that p-values were calculated using ANOVA or t-tests.
- The authors conducted a post-hoc analysis to calculate 95% confidence intervals in Table 2, but since Table 3 presents regression analysis results that closely mirror those in Table 2, the post-hoc analysis in Table 2 can be omitted for parsimony. However, I leave this decision to the author.
- Table 3: Instead of reporting p < 0.01, please provide exact three-decimal p-values.
Discussion
The Discussion and Conclusion sections are well-written and appropriately acknowledge the potential bidirectional relationships.
- The non-significant association between living alone and LSIS-6 is unexpected. Why might this relationship differ from findings in Western contexts? Please consider adding a brief discussion or hypothesis on this point (Lines 271–275).
Author Response
Comments 1: Thank you for inviting me to review this paper. This cross-sectional study examines the factors associated with loneliness and social isolation among Korean adults. Given the growing social concerns on loneliness and social isolation both in Korea and globally, this study is timely and valuable. This paper is very well-written and based on a sound methodology. Particularly, the use of a nationally representative sample and a validated multi-item measurement (LSIS-6) enhances the generalizability of the findings. I appreciate the author' efforts and would like to offer a few minor suggestions for improvement.
Response 1: Thank you for taking the time to review the manuscript and the positive feedback.
Methods
Comments 2: The Method section provides a clear and reasonable explanation of the sampling procedure, measurement, and statistical analysis.
Section 2.1: It would be helpful to briefly describe the COVID-19 situation in Korea between June and August 2021, along with the relevant social distancing policies, to provide context.
Response 2: I added the following phrase, “while social distancing measures were still in place during the COVID-19 pandemic.” (lines 111-112)
Comments 3: Lines 190–191: How was collinearity checked? If the variance inflation factor (VIF) was used, please specify its cutoff value for ruling out multicollinearity.
Response 3: I used tolerance to check for collinearity. Tolerance values less than 0.10 indicate collinearity. I confirm all variables have tolerance values of greater than 0.10. This point is added in the Methods-statistical analysis section. (line 212)
Results
Comments 4: Table 2: Please include a table footnote indicating that p-values were calculated using ANOVA or t-tests.
Response 4: I included a footnote to the Table 2 indicating that p-values were calculated using ANOVA or t-tests.
Comments 5: The authors conducted a post-hoc analysis to calculate 95% confidence intervals in Table 2, but since Table 3 presents regression analysis results that closely mirror those in Table 2, the post-hoc analysis in Table 2 can be omitted for parsimony. However, I leave this decision to the author.
Response 5: I appreciate and understand the comment; however, I have decided to keep Table 2 as it is, as long as it does not negatively impact the paper.
Comments 6: Table 3: Instead of reporting p < 0.01, please provide exact three-decimal p-values.
Response 6: I changed all p-values to three decimal places in Table 3.
Discussion
Comments 7: The Discussion and Conclusion sections are well-written and appropriately acknowledge the potential bidirectional relationships.
The non-significant association between living alone and LSIS-6 is unexpected. Why might this relationship differ from findings in Western contexts? Please consider adding a brief discussion or hypothesis on this point (Lines 271–275).
Response 7: I added the followings in the fourth paragraph of the discussion (lines 302-305): One possible explanation for this difference is that, unlike in some other high-income countries, most people in Korea live in high-rise apartments in densely populated urban areas. Consequently, living alone does not necessarily equate to physical or social isolation.
Round 2
Reviewer 1 Report
Comments and Suggestions for Authors
Overall, this revised manuscript is improved. Questions I raised were addressed and explained in the text that clarified the original issues. I believe that the findings in this study are supported by the analysis though I still find that the significance of the findings are still a bit unclear. One issue is that it makes more intuitive sense for loneliness to predict mental health outcomes (depressive symptoms, anxiety) than the other way around. That said, a little more justification for why loneliness is the outcome rather than mental health is warranted in the introduction.
In terms of the discussion, I think the paper would benefit from being more precise regarding the policy implications. For example, which groups should be targeted for interventions on loneliness and isolation prevention? Given the large number of men and the low fertility rates in Korea, it seems a bit broad to target those groups for a socialization intervention. More concrete examples would strengthen those implications.
As a minor note, the table title for the regression results should include the dependent variable to ensure clarity for readers.
Another note, the references are still out of alphabetic order.
Author Response
Comments 1: Overall, this revised manuscript is improved. Questions I raised were addressed and explained in the text that clarified the original issues. I believe that the findings in this study are supported by the analysis though I still find that the significance of the findings are still a bit unclear. One issue is that it makes more intuitive sense for loneliness to predict mental health outcomes (depressive symptoms, anxiety) than the other way around. That said, a little more justification for why loneliness is the outcome rather than mental health is warranted in the introduction.
Response 1: Thank you for taking the time to review the revised paper and provide feedback. I understand that loneliness can be a risk factor rather than an outcome. However, the literature regards the issue of loneliness as a state that has to be estimated and explained. The ambiguity actually led me to include 'correlates' in the title of the paper. In the absence of a more compelling rationale, I added the following phrase in the first paragraph of the introduction section: '… and regarded as a state that has been critically assessed.' (lines 39-40)
Comments 2: In terms of the discussion, I think the paper would benefit from being more precise regarding the policy implications. For example, which groups should be targeted for interventions on loneliness and isolation prevention? Given the large number of men and the low fertility rates in Korea, it seems a bit broad to target those groups for a socialization intervention. More concrete examples would strengthen those implications.
Response 2: I revised the last paragraph of the discussion section to include the phrase: “particularly those with multiple chronic conditions and/or health behavior issues related to nicotine and alcohol use.” (lines 365-367).
Comments 3: As a minor note, the table title for the regression results should include the dependent variable to ensure clarity for readers.
Response 3: I changed the title of Table 3 “Results of multiple regression analyses on the 6-item Loneliness and Social Isolation Scale score” as advised.
Comments 4: Another note, the references are still out of alphabetic order.
Response 4: I checked the style format required for Healthcare and other papers published in the journal. References appear in the order they appear in the text, which is what my paper did. Please enlighten me for what to do.